# Fast topographic optical imaging using encoded search focal scan

Narcís Vilar [1,2], Roger Artigas[1], Martí Duocastella [2] ✉ & Guillem Carles [1] ✉

A central quest in optics is to rapidly extract quantitative information from a sample. Existing topographical imaging tools allow non-contact and three-dimensional measurements at the micro and nanoscales and are essential in applications including precision engineering and optical quality control. However, these techniques involve acquiring a focal stack of images, a time-consuming process that prevents measurement of moving samples. Here, we propose a method for increasing the speed of topographic imaging by orders of magnitude. Our approach involves collecting a reduced set of images, each integrated during the full focal scan, whilst the illumination is synchronously modulated during exposure. By properly designing the modulation sequence for each image, unambiguous reconstruction of the object height map is achieved using far fewer images than conventional methods. We describe the theoretical foundations of our technique, characterise its performance, and demonstrate sub-micrometric topographic imaging over 100 μm range of static and dynamic systems at rates as high as 67 topographies per second, limited by the camera frame rate. The high speed of the technique and its ease of implementation could enable a paradigm shift in optical metrology, allowing the real-time characterisation of large or rapidly moving samples.

Topographic optical imaging at the microscale plays a crucial role in industrial and scientific processes[1,2]. Examples include optical inspection in production lines[3], metrology of additively manufactured parts[4], and three-dimensional (3D) surface measurement of biomaterials[5]. Typically, topography maps are reconstructed from a z-stack, that is, a collection of images from different focal planes acquired sequentially, as in confocal microscopy[6] or interferometry[7]. This process can be slow, in particular for bulky objects that require large mechanical displacements between focus and object. The problem can be further aggravated when pursuing sub-micrometric optical resolution. In this case, high numerical aperture (NA) optics are needed which feature a short depth-of-field, and consequently, require a large number of z-planes for proper reconstruction. As a result, commercial surface microscopes are ill-suited for characterising fast-moving industrial processes or rapidly evolving biological systems.

Several strategies have been developed to reduce acquisition time in topographic imaging of bulky objects. They can be broadly classified into two main groups. The first one consists of techniques aimed at speeding up the assembly of a z-stack. Examples that fall into this category include spatiotemporal multiplexing[8], multifocus microscopy[9,10], encoded illumination methods[11–13], light-field imaging[14], or using variable optical elements for fast focus control[15]. Whilst capable of retrieving 3D information from a sample in real-time, these new approaches can be limited in axial range, ease of use, cost of implementation, or the range of samples they can characterise. Alternatively, it is possible to use a second group of techniques based on reducing the information necessary for 3D image reconstruction. They achieve so by exploiting the intrinsic sparsity of most common samples. This can be as simple as selecting predefined regions in a sample, as in random scanning microscopy[16], but the required a priori information is not generally accessible. Instead, computational approaches such as compressed sensing or point-spread function engineering[17,18] enable full topographic reconstruction from far fewer images – up to one order of magnitude – than traditional z-stacks[19]. Still, the gain in 3D

¹Sensofar Tech S.L, Parc audiovisual de Catalunya, BV-1274 km 1, 08225 Terrassa, Spain. ²Department of Applied Physics, Universitat de Barcelona, C/Martí i Franquès 1, 08028 Barcelona, Spain. ✉e-mail: marti.duocastella@ub.edu; gcarles@sensofar.com

imaging speed can come at the cost of reconstruction fidelity. Also, these techniques are computationally expensive, typically requiring offline processing[20]. Simply put, a simple-to-implement technique for real-time topographic optical imaging of large volumes at the microscale does not exist.

Here, we introduce Encoded Search Focal Scan (ESFS), a technique that fills this void and enables real-time reconstruction of topography maps at micro and nano scale resolution. Our strategy is based on a different scanning paradigm that does not follow the traditional sequential plane-by-plane scan. We use a reduced set of images, each acquired during a complete focal sweep through the axial measurement range of interest. By using synchronised pulsed illumination during the scan, we can encode information about the axial location of the sample. As a result, the strong data sparsity inherent in three-dimensional topographic imaging can be exploited, enabling order-of-magnitude improvements in acquisition time, only limited by camera frame rate and axial sweeping time, without added computational complexity or sacrifice in reconstruction fidelity. Our method can be implemented in any confocal-like system with minor modifications. We report here experimental demonstration of ESFS for fast 3D topographic imaging, including the reconstruction of surface topographies spanning 100 μm in the axial range from only eight images with a precision below 50 nm, and the 3D capture of the temporal deformation of a micro-electromechanical system (MEMS) gas sensor in real-time. These are examples of 3D topographic imaging at speeds not possible before.

## Results

### Working principle of the Encoded Search Focal Scan

The salient aim of topographic imaging at the microscale is to determine the axial position of an object. For this, conventional approaches – confocal-like or focus variation systems – scan the axial range where we know the sample lies into $N$ planes and interrogate all planes querying the presence of the object. Because each plane corresponds to an image, that would require $N$ images. Note that $N$ depends on the a priori information of our sample, the properties of the optical system, and the scanning range, but it is typically a large number. However, we can optimise sample search provided the sample is sparse, that is, it is thin enough to be assigned to a unique position within the z-scanned range. In this case, we can perform a binary search, in which we interrogate merged groups of planes and check whether our sample is or is not in each group. This is the core idea of ESFS, which can lead to a dramatic reduction in the number of queries, and therefore of images. Indeed, by properly selecting the merged interrogation planes, a binary search is possible, enabling to determine the axial position of the sample using only $\log_2(N)$ images.

The gist of the method presented here is to implement such binary search in an optical microscope. To this end, we acquire a set of images, each captured whilst the optical focus is axially scanned through the entire measurement range, as illustrated in Fig. 1(a). Information from the in-focus plane and all other planes is thus merged onto each image. The axial location of the in-focus plane is unknown, as it depends on the height of the sample, and constitutes our measurand. During the focal sweep, the illumination is turned on and off with a precisely controlled sequence that is different and unique for each image. We denote a particular illumination sequence $M_i(z)$ where $z$ is the continuous axial coordinate, such that $M_i(z) = 0$ implies the illumination is off at plane $z$ and $M_i(z) = 1$ implies the illumination is on. Therefore, setting $M_i(z)$ as an on-off sequence, effectively determines the groups of merged interrogation planes. For a given $M_i(z)$, the corresponding acquired image can be written as,

$$I_i(x,y) = \int_{z_{min}}^{z_{max}} \left\{ \left[ r(x,y) \cdot L(x,y) \right] * \text{PSF}\left(x,y,z-z_s\right) \right\} M_i(z)\, dz, \quad (1)$$

where $z_{max} - z_{min}$ is the measurement range, $r(x,y)$ is the reflectivity of the sample, $L(x,y)$ is the illumination of the sample plane, PSF is the imaging point-spread function, $z_s$ is the height of the sample at location $(x,y)$ that we intend to measure, the operator $*$ denotes two-dimensional convolution in the tangential plane $(x, y)$, and index $i$ denotes the illumination sequence. We assume the focal sweep is implemented in a constant velocity $v$ through the measurement range and therefore time and $z$ can be interchanged through $z = vt$, but any other relation $z(t)$ would be applicable as long as it is known and monotonic. Note that it is not possible to infer the axial location of the sample from a single focal-sweep image. However, for any sequence $M_i$, it is possible to detect whether the illumination was on or off at the time the sample was in-focus during the focal sweep. For this, all we need is to calculate a signal from the image that is focus-sensitive.

Such focus-sensitive signal can be obtained, for instance, in confocal systems by using a pinhole[21] or in structured illumination microscopy[22], but in-plane scanning or multiple images per plane are needed, respectively. Alternatively, it is possible to achieve a focus-sensitive signal by simply projecting a static illumination pattern on the sample and analysing the high spatial-frequency content from a single captured image – without in-plane scanning. The origin of such focus sensitivity lies in the blurring effect of defocus: during the integration through the focal sweep, all spatial detail is blurred out for all planes except for the region close to the sample location. Therefore, by querying for high-frequency detail, we probe if the sample is or is not in any of the illuminated axial locations defined by $M_i$. In other words, in the axial convolution in Eq. (1), $\text{PSF}(x,y,z-z_s)$ acts as a strong low-pass filter due to defocus if $z \neq z_s$, and so the high spatial-frequency features are only recorded if $M_i(z = z_s) \neq 0$ (within the depth of field). For instance, a metric based on the magnitude of the Laplacian of the image field provides such signal, as shown in Fig. 1:

$$S_i(x,y) = \mathscr{G}_{\sigma_b}\left\{ \left| \nabla^2 \mathscr{G}_{\sigma_a}\left\{ I_i(x,y) \right\} \right| \right\} \quad (2)$$

where $\mathscr{G}_\sigma\{\cdot\}$ denotes spatial Gaussian filtering with standard deviation $\sigma$, included for suppressing noise and smoothing. It is this information that is exploited in ESFS: through a simple threshold on $S_i(x,y)$, we detect the presence or absence of the sample at any of the locations masked axially by the binary modulation $M_i(z)$. Note how this calculation of the focus-sensitive signal involves spatial filtering on a single image and, although this affects the lateral resolution to some extent, it does not require in-plane scanning.

A simple example of the information extracted from a focal-sweep image is the implementation $M_i(z) = 1$ for $z < (z_{max} - z_{min})/2$ and $M_i(z) = 0$ otherwise. That is, the illumination would be on during the first half of the exposure time and off during the second half, making the fixed structured illumination pattern only visible if the sample is in-focus at some point during the first half of the exposure time. Therefore, in this example sequence, a value of $S_i(x,y)$ above (or below) the set threshold would indicate that the height of the sample at $(x,y)$ is somewhere in the first (or second) half of the measurement range. The combination of a few images with an appropriately designed set of sequences $\{M_i(z)\}$ can be used to perform an axial search of the height at each pixel (see Supplementary Note 1). That is, if the measurement range is divided into $N$ steps of size $T = (z_{max} - z_{min})/N$, we can find the step number, $d$, that corresponds with the axial location of the sample at each pixel. Note how such a task would require $N$ images, one per step, in a conventional plane-by-plane scan, as illustrated in Fig. 1b. Instead of such an exhaustive search, ESFS implements a binary search[23] in which only $\log_2(N)$ images are required to perform the search, as illustrated in Fig. 1d. This measuring paradigm leads to a great reduction in the number of necessary images. For instance, to sample a measurement range of 400 μm at steps of 1.6 μm, a conventional plane-by-plane scan would require 250 images, whereas ESFS would provide equivalent axial localisation with only 8 images.

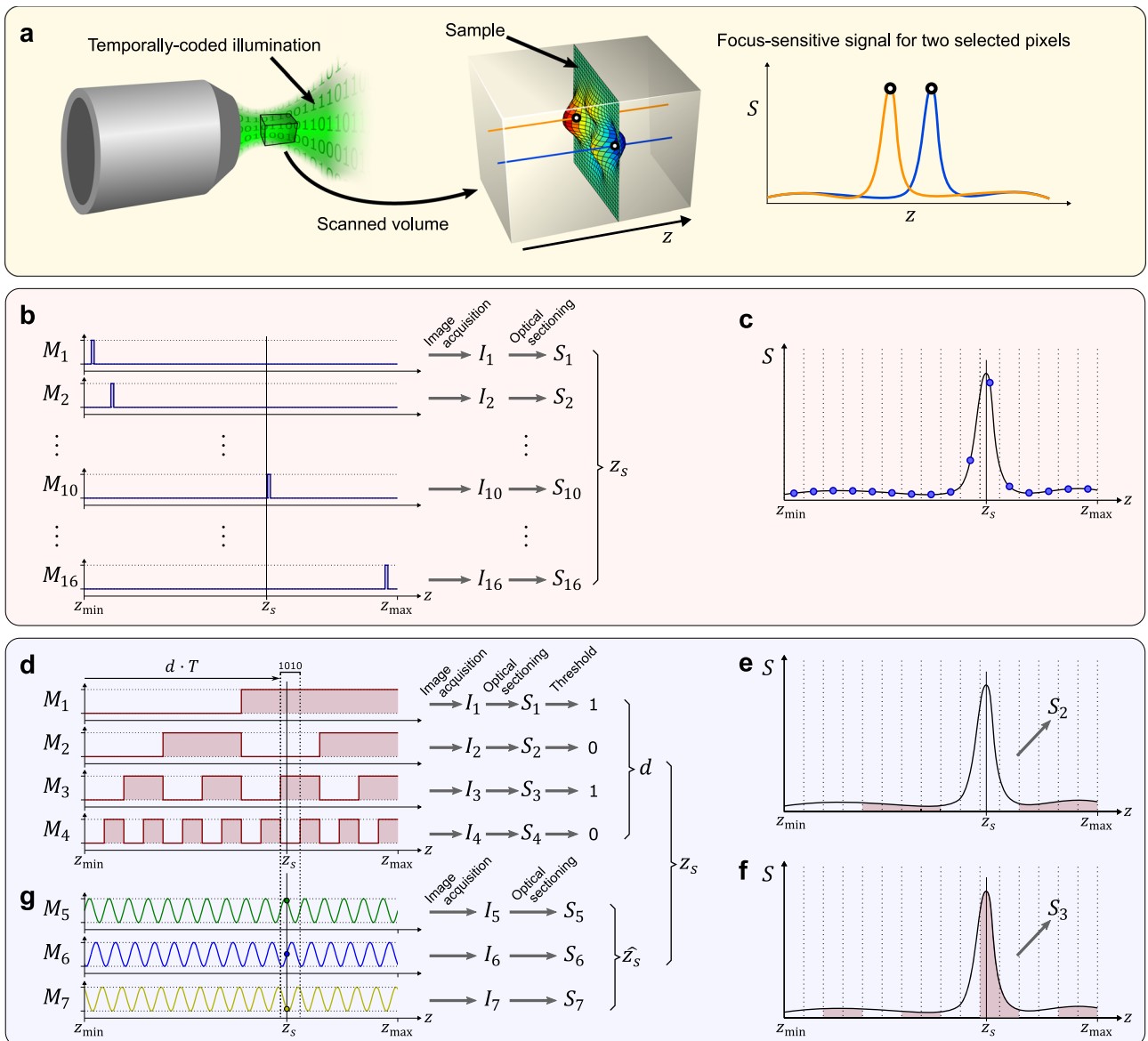

**Fig. 1 | Overview of the ESFS method.** A volume of interest is defined by the field of view of the imaging system and the axial measurement range, containing the sample to be measured, as illustrated in (**a**); the signal from Eq. (2) is plotted at the right graph for two selected pixels across the measurement range. Conventional scanning (**b**) samples the signal with a uniform sampling rate as illustrated in (**c**), requiring a large number of samples/images. In the ESFS method, information is extracted with far fewer samples/images. In the first stage (**d**), a set of images $I_i$ with coding sequences $M_i$ are acquired. Binarisation of the focus-sensitive signal, $S_i$, provides the code to calculate the step number, $d$, that contains the height of the sample, $z_s$. Illustration of the integrated signals $S_2$ and $S_3$ are shown in (**e**) and (**f**), respectively, to highlight how they respond to the presence of the sample. Note the illustration is shown for one pixel with height $z_s$ but results are calculated in parallel for all pixels. In a second stage (**g**), additional images are acquired with a time-modulated intensity illumination; in this example three images are acquired with sinusoidal modulation and with $2\pi/3$ relative phase shift. The relative strength of the focus-sensitive signal can be used to calculate the axial location of the sample with increased accuracy, albeit modulo the step size, $T$. Results from the first stage are then used to unambiguously unwrap the measurement.

In all, if the encoding sequences $M_i$ implement a basic binary code, then the step number that contains the axial location of the object at any lateral location within the captured field of view may be inferred by,

$$d(x,y) = \sum_{i=1}^{n} 2^{n-i} H\big(S_i(x,y) - S_{th}(x,y)\big) \quad (3)$$

where $n$ is the number of sequences/images, $H(\cdot)$ is the Heaviside step function used to binarise the field $S_i(x,y)$, and $S_{th}(x,y)$ is a calibrated threshold over which the object is deemed detected (see Supplementary Information for details). For any other coding sequence, such as a Gray code[24], a decoding step precedes the determination of the step number.

The minimum height of the axial step that can be meaningfully implemented in ESFS is dictated by the depth-of-field of the objective, as it determines the ability to resolve axially the detection of the sample. Implementing smaller steps would not yield increased axial resolution. However, it is possible to further exploit the prior knowledge of sparsity to maximise localisation precision. In conventional approaches this is typically done by fitting a curve to the axial response[25]. Likewise, in ESFS it is possible to further increase the axial resolution. To this end, we propose and implement an additional stage in ESFS. In this second stage, we acquire a few more focal-sweep

images, modulating the intensity of the illumination in time. One option is to implement a sinusoidal modulation with a period corresponding to the time associated with a single axial step in the previous stage. In other words, we acquire a focal-sweep image with an illumination intensity modulated sinusoidally in time and with $N$ cycles through the measurement range, as illustrated by the example in Fig. 1g. That is, we set the modulation at stage two as,

$$M_j(z) = \frac{1 + \cos\left(\frac{2\pi}{T} z + \delta_j\right)}{2} \tag{4}$$

where $\delta_j$ is a phase offset, and $T$ is the step size defined in the previous stage satisfying $TN = z_{\max} - z_{\min}$. Because the filtering in Eq. (2) extracts high spatial frequency information, which is only present at planes close to $z_s$, the magnitude of $S_j(x, y)$ is proportional to $M_j(z = z_s)$, and so we can write,

$$S_j(x, y) = S_{\max}(x, y) \frac{1 + m \cos\left(\frac{2\pi}{T} z_s + \delta_j\right)}{2} \tag{5}$$

where $S_{\max}(x, y)$ is the value of the signal in Eq. (2) when $M_j(z) = 1$, and $m$ is the resulting reduction in contrast that appears due to the depth of field of the imaging system. Repeating the measurement with different phase shift $\delta_j$, it is possible to solve for the axial location of the sample. As an example, implementing a set of phase shifts $\delta_j = \{-2\pi/3, 0, 2\pi/3\}$, the axial location of the sample can be readily computed as,

$$\hat{z}_s(x, y) = \frac{T}{2\pi} \arg\left(\sqrt{3}(S_1 - S_3) + \imath(2S_2 - S_1 - S_3)\right) \tag{6}$$

where $\arg(\cdot)$ is the argument of a complex number in the range $[0, 2\pi)$ and $\imath = \sqrt{-1}$.

Although originated from entirely different physical processes, the described formalism is equivalent to that of phase-shifting interferometry (PSI). In fact, Eq. (6) is the well-known Three Step Algorithm[26]. There is a plethora of algorithms and variations that work with a different number of images and relative phase shifts in PSI that would be equally applicable to ESFS. As is also the case in PSI, the result of Eq. (6) is the topography of the sample wrapped at the steps of our illumination cycle. Note that such wrapping restricts PSI and similar techniques to shallow samples. Importantly, though, results from the first stage in ESFS provide a mechanism for performing a robust and unambiguous unwrapping, as,

$$z_s(x, y) = z_{\min} + d \cdot T + \hat{z}_s(x, y) \tag{7}$$

where $d$ is the step number obtained in the first stage. Therefore, by combining the two illumination stages, ESFS maximises localisation precision whilst maintaining a large axial range and a highly reduced set of acquired images (see Supplementary Information).

## Implementation

ESFS can be readily implemented in any optical system for topographic imaging. The required main elements are systems for light modulation, structured illumination, and focal scanning. We report here two ESFS prototypes. They both feature a pulsed Light Emitting Diode (LED) for the light modulation, and a fixed patterned amplitude mask to generate the structured illumination, but employ a different focal scanning mechanism.

The first prototype includes a motorised stage to perform the focal sweep by moving the microscope objective relative to the sample and is shown in Fig. 2. The system specifications and design are as follows. The measurement range is set to $z_R = 100\,\mu m$, the motorised stage is configured to move at a constant velocity of $1000\,\mu m/s$, and

therefore we set the camera exposure time at 100 ms so that the sample is moved through the entire measurement range during exposure. We employ a 20x/0.45NA objective, featuring a depth-of-field of approximately 2.6 µm. We decided to divide the measurement range using $N = 8$ steps, with a step size of $T = 12.5\,\mu m$, requiring only 3 binary-modulated images. Note that the measurement range is approximately 40 times the depth-of-field, so the maximum number of binary-modulated images would be 6 (as is the lowest integer satisfying $2^n > 40$). However, for the sake of faster operation, we used fewer images and larger steps. An analysis of how this affects performance is included in Supplementary Note 2. For the second stage, we implement pulsed illumination with a matching period of 12.5 ms, and acquire 4 images with equally-distributed phase shifts (in practice, implementing pulsed illumination with period $T$ provides virtually identical results as implementing sinusoidal modulation, see Supplementary Note 1). Additionally, for a robust simultaneous determination of the step number $d$ and the wrapped measurement $\hat{z}_s$, we implement a Gray code using an additional image in the first ESFS stage, corresponding to the higher spatial frequency coding (see Supplementary Note 1 for details). In all, we acquire a total of 8 images, 4 images in the first stage and 4 in the second stage.

Results for the first prototype are shown in Fig. 2. In Fig. 2b it is shown the measurement of a certified step height sample. In this case, we retrieved a height value of 21.704 µm, which is in excellent agreement with the certified value for the specimen, of 21.702 µm ± 0.022 µm. In Fig. 2c it is shown the topography map of a standard roughness sample AIRB40 from NPL[27], with a measured surface roughness parameters[28] of Sa = 0.77 µm and Sq = 0.99 µm; and its certified values are Sa = 0.79 µm with an expanded uncertainty of 0.03 µm and Sq = 1.00 µm with an expanded uncertainty of 0.02 µm, again with excellent agreement. And two examples of the relief from two regions of a coin are shown in Fig. 2d, e. In all cases, the measurement range is 100 µm and reconstructed topographic maps were built from only eight acquired images. Further details of the characterisation of the system can be found in Supplementary Information, including a comparative analysis of the topographic spatial resolution and system noise achieved with ESFS and with a conventional technique based on plane-by-plane scanning. Results show that ESFS leads to increased system noise but causes only a slight reduction in the lateral resolution.

The second prototype that we report here is implemented on a commercial microscope, modified to include a custom-made illumination system and a fast varifocal tuneable acoustic gradient (TAG) lens, as shown in Fig. 3a. When placed at a conjugate plane of the back pupil of the objective using a 4 f system, the TAG lens allows for sinusoidal focal sweeping at microsecond time scales. Thus, by using pulsed illumination synchronised with the TAG lens driving signal, we achieved ESFS at speeds only limited by the camera frame rate. Initially, we calibrated the response of the TAG-enabled ESFS system for a given scan range. In all cases, we employed a 20x/0.45NA objective and worked within the linear region of the sinusoidal axial scan. At these conditions, the calibration simply consists of a scale factor, which is given by the illumination period, $T$ (Eq. (6)). To determine its value, we measured 3 different certified step height samples and performed a linear fit, as shown in Fig. 3b.

Once the system is calibrated, we reconstructed ESFS topographies of step height samples and flat mirrors, as shown in Fig. 3c-e. The ESFS parameters were the same as in the first prototype regarding the number of steps (8) and images (4 for the first stage, 4 for the second), but the resulting measurement axial range was approximately 54 µm. Notably, the measurement time was only limited by the camera frame rate – 539 frames per second in current experiments –, thus one topography could be obtained in only 15 ms. This represents a remarkable speed of 67 topographies per second.

To demonstrate the benefit of fast ESFS-enabled topographic imaging, we first measured the three-dimensional profiles of a rapidly

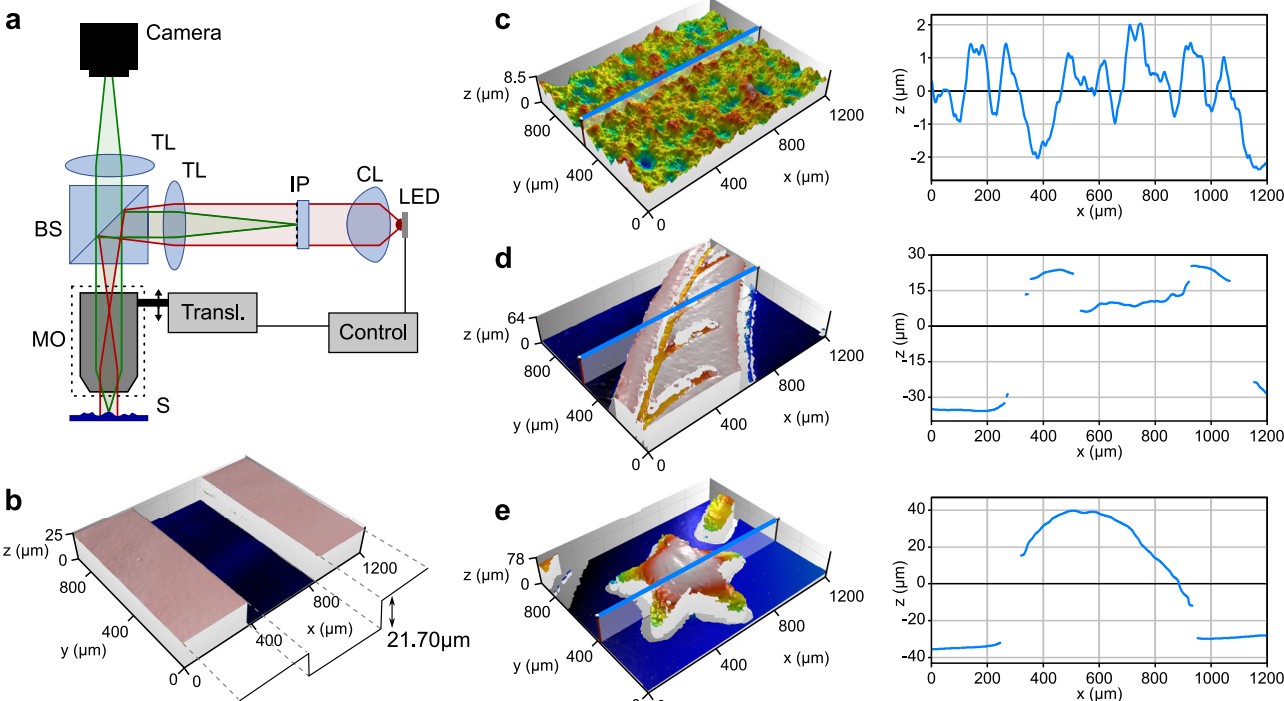

**Fig. 2 | Implementation of the ESFS method in an optical system. a** Optical layout including a microscope objective (MO), beam splitter (BS), tube lens (TL), camera, illumination system including LED, collimating lens (CL), and image plane pattern (IP) projected at the sample focal plane; an electronics control synchronises the illumination and the scanning with the acquisition, and the scanning is performed through a translation stage that moves the objective. **b–e** Reconstruction topographic results using the ESFS system: (**b**) step height measurement; (**c**) measurement of rugosity standard AIRB40; and (**d, e**) measurement on regions of a coin. Height profiles across the field of view (in the direction shown in the associated topographies) are depicted in the right graphs of (**c–e**). In some regions of the measurements shown in (**d, e**) the sample has a local slope that exceeds the NA of the objective, light reflected off the sample is not captured by the objective, and they appear as non-measured regions.

moving sample. In more detail, we used the roughness standard AIRB40 from NPL[27] mounted on a translation stage and manually translated the sample whilst imaging at 539 frames per second. Results are shown in Supplementary Video 1, and a few selected frames are shown in Fig. 4a. In this case, as the sample is translated through the field of view of the system, we can seamlessly stitch the topographies frame by frame, as is shown in the figure. This highlights how the fast operation of ESFS enables measuring moving samples that are difficult or impossible to characterise with conventional topographic optical systems. They include samples that cannot be properly fixed and are prompt to vibrations, or samples that are in constant movement, as occurs during the optical inspection in production lines. In these cases, ESFS enables to increase the rate of inspected parts, effectively reducing the cost of inspection per part. To further assess the benefit of ESFS, we imaged a functional and dynamic MEMS device, as shown in Fig. 4a. Specifically, we measured the temporal deformation of the suspended membrane of a micromachined gas sensor[29], working in modulated operation mode. The suspended micro-hotplate is heated through an electrical current that varies sinusoidally in time. Such a modulation allows performing a spectral analysis of the gas characteristics and it can help to avoid drift effects caused by changes in the temperature of the environment[30]. However, the suspended membrane can suffer from mechanical deformations due to temperature-induced stress, ultimately limiting the gas-sensing performance of the device. ESFS enables video-rate inspection (67 topographies per second) of these deformations with good resolution in all three dimensions of space. Results from such measurement using a 6 Hz modulated driving electrical signal are shown in Supplementary Video 2, and selected frames are shown in Fig. 4b as an illustration. Notably, the periodic changes in the membrane height, of about 2 μm, can be clearly resolved. More importantly, the high speed of ESFS also allows characterising membranes modulated at even higher rates (up to 20 Hz, Supplementary Video 3). Therefore, the remarkable speed and sub-micrometric spatial resolution of our system deem it appropriate for characterising dynamic samples or rapid processes.

## Discussion

The ESFS technique enables to reconstruct topographic manps of 3D samples with orders of magnitude reduction in the number of required input images. It consists of two different stages. A first, so-called binary stage, that coarsely localises the sample within the measurement range; and a second, so-called phase-shifting stage that localises the sample with higher precision. This two-step process resembles that used in fringe projection profilometry (FPP) to increase topographic imaging speed[31]. In this case, different patterns of structured light are sequentially projected at the sample, from which the sample axial position can be inferred by triangulation – no axial scanning is needed. However, a measurement range exceeding the depth-of-field of the detection optics impacts the axial and, severely, the lateral resolutions due to the defocus blurring. Coupled with the geometry constraints of angled projection and limitations of triangulation, FPP typically features a spatial resolution of hundreds of microns, leaving the technique out of the domain of high-resolution microscopy[32]. Such resolution can be orders of magnitude worse than what is possible using ESFS, where the use of full NA is enabled, and the measurement range can be arbitrarily extended.

As with any measurement technique, motion artifacts would be expected if one attempts to capture dynamic processes that change faster than the sampling rate. However, the high reduction in the number of input images increases the sampling rate, effectively rendering ESFS as a real-time technique. For instance, in our implementation in the second prototype data acquisition time for one

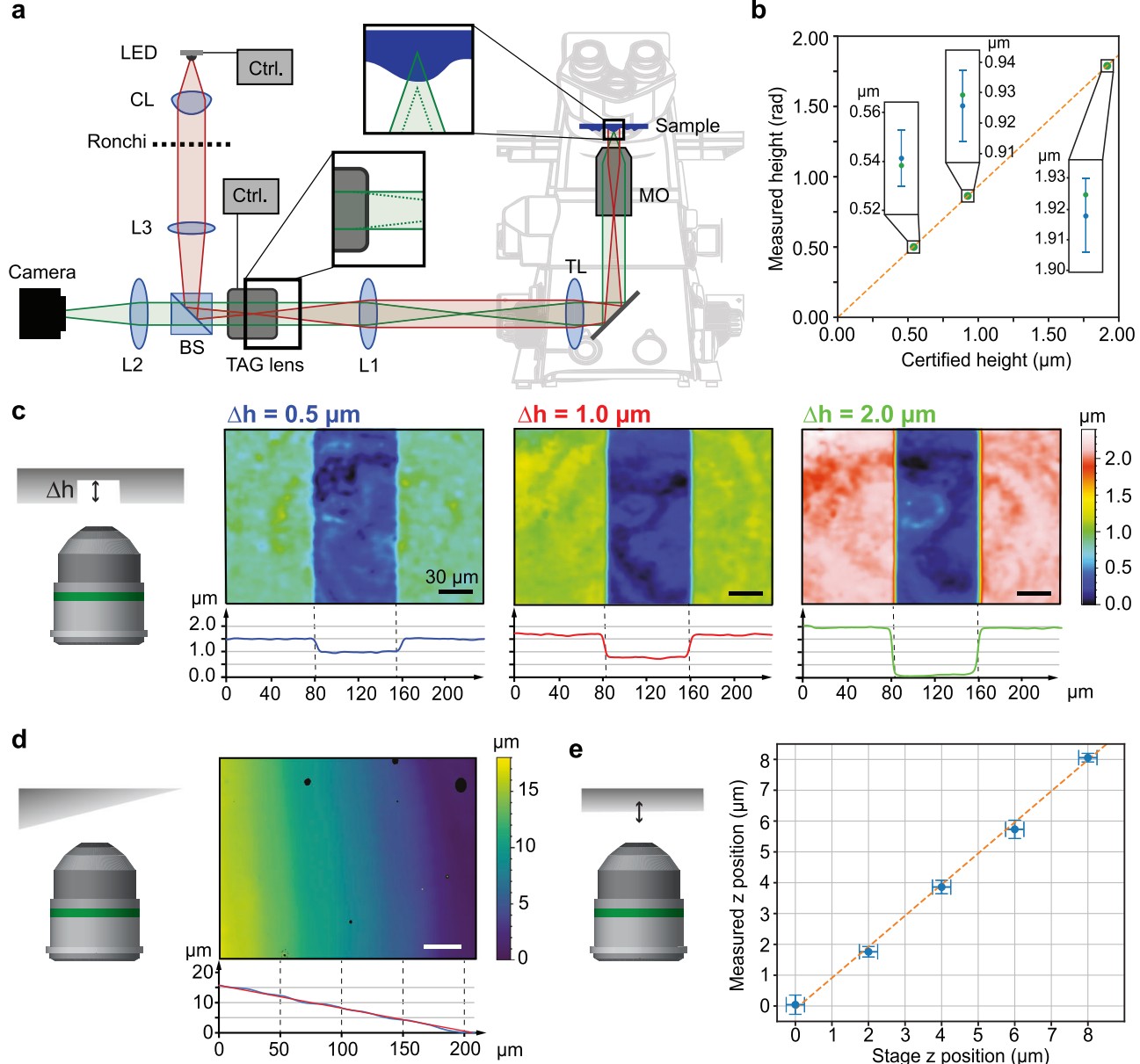

**Fig. 3 | Implementation of the ESFS method in a commercial microscope with a fast varifocal lens. a** Scheme of the optical layout including a microscope objective (MO), beam splitter (BS), tube lens (TL), TAG lens, relay lens system (L1 and L2), camera, illumination system including LED, collimating lens (CL), Ronchi pattern and lens (L3); an electronics control synchronises the illumination and the TAG lens driving signal. **b** Plot of the measured height, in radians, versus the certified height used to calibrate the system. The blue dots depicted in the inset graphs correspond to the certified values; the fitted calibration factor was used to convert phase to heights in the inset graphs. **c–e** Reconstruction topographic results using the ESFS system: **c** Colormaps of three step height measurements of 0.5, 1.0, and 2.0 μm. The average height profile for each colormap is shown at the bottom. **d** Colormap of a tilted mirror, and corresponding height plot (bottom). The red solid line corresponds to the ground-truth value. **e** Plot of the measured axial position of a flat mirror as a function of its axial location, controlled using a z stage. The dashed orange line corresponds to a line with slope 1.

measurement was only 15 ms, enabling imaging at 67 topographies per second. Furthermore, reconstruction in ESFS is not a computationally intensive operation, readily enabling online processing at real-time, see Methods.

In the form elaborated in this paper, the optical sectioning signal is calculated after the focal-sweep images have been acquired. This leads to a trade-off: increasing the measurement range reduces the strength of the optical sectioning signal, yielding a lower signal-to-noise ratio (SNR). An analysis of how this affects the measurement precision is included in Supplementary Note 2. This issue would be avoided if ESFS is implemented such that the focus-sensitive signal is detected directly, for example, in a confocal microscope. In this case, out-of-focus light is not detected, and a useful SNR over an arbitrarily large measurement range should be possible. Alternatively, even with an indirect computation of the optically sectioned images, it would be possible to maintain a high SNR over large ranges provided the sample is shallow. This may be the case, for example, for fast inspection of parts in roll-to-roll processing. In these situations, the output results from the ESFS first stage, may be used as input for an optimised acquisition of the second stage images. By activating the illumination only at the axial range containing the sample height histogram, SNR in the second-stage images would not be reduced even if we increase the measurement range. This could be done in two ways: modifying the focal scanning system so that the focal sweep only covers the axial range of the sample, or simply deactivating the illumination for all axial locations that are not within the axial range of the sample. In any case,

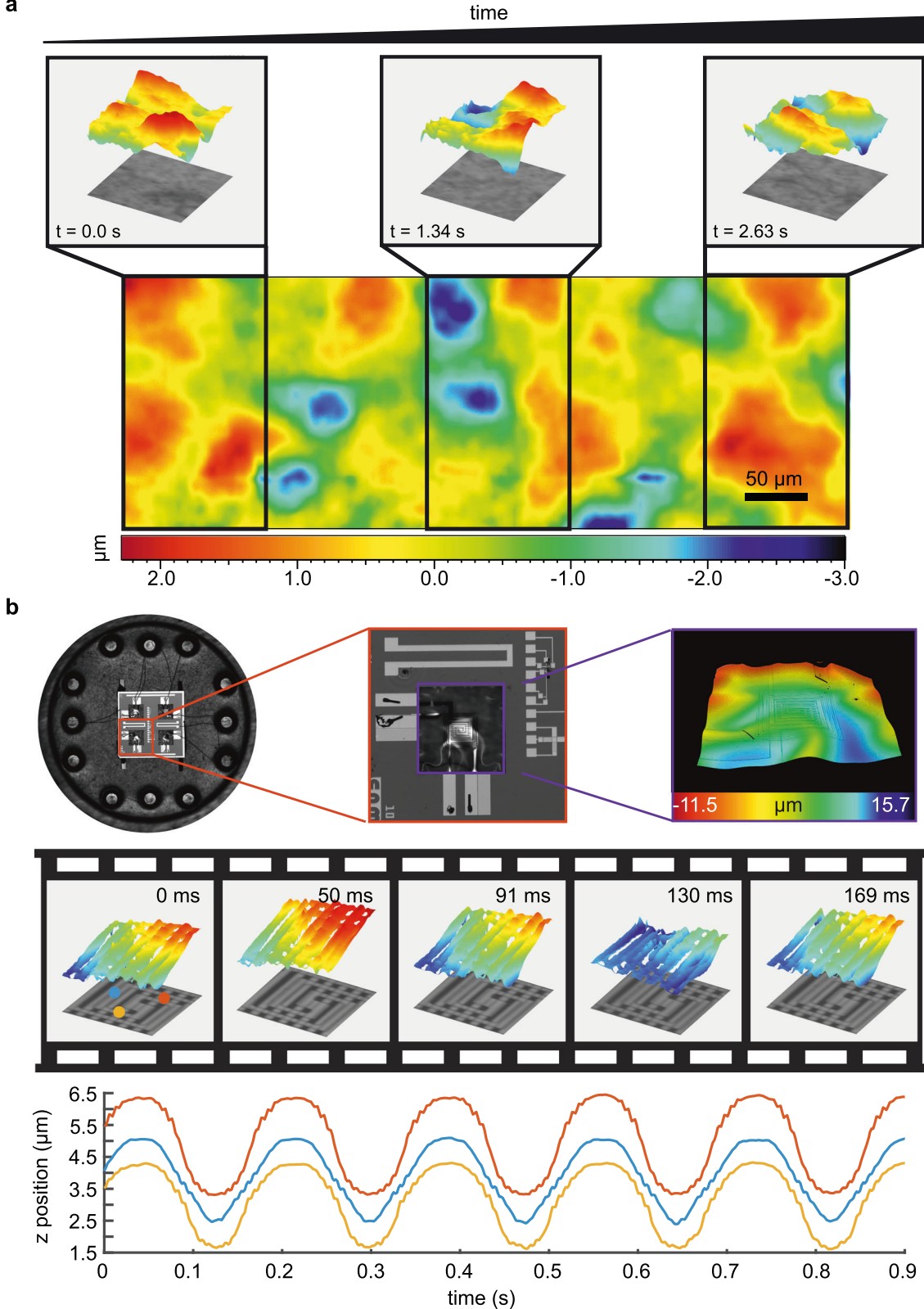

**Fig. 4 | Topographic reconstruction of dynamic samples. a** Stitching of 3D profiles acquired with ESFS at 67 topographies per second during the manual translation of the roughness standard AIRB40. The insets correspond to topographic images acquired at the indicated time instances, with the intensity image underlaid. **b** Top: optical micrograph of 4 MEMS gas sensors assembled into a chip with the corresponding connectors (left); optical micrograph of an individual MEMS gas sensor consisting of an interdigitated circuit and a suspended membrane (centre); 3D optical profile of the membrane and circuit (right), imaged area 844.56 × 706.56 μm. Centre: ESFS-enabled topographic reconstruction, with the intensity image underlaid, at different time instances of the membrane deformation caused by periodic electric-induced heating at 6 Hz. Imaged volume 248.4 × 186.3 × 54 μm³. Bottom: plot of the axial position overtime at 3 different positions within the membrane. The data was extracted from the ESFS reconstructions acquired at 67 topographies per second.

the measurement range would be safely extended without dropping performance.

Implementation of the ESFS first stage using a binary code, as we described, has the advantage that calculation of the step number can be done by application of a simple threshold to the optically sectioned images, and also that is robust with respect to lateral variations in the reflectivity of the sample. However, since the reflectivity map can be calibrated (see Supplementary Note 1), it would be straightforward to implement ESFS with a tertiary code using a two-threshold value. And in fact, only SNR in the images limits the practical implementation of codes with higher bases, which would allow further reducing the number of images required to localise the sample within the measurement range.

We believe that the unique combination of speed and spatial resolution of ESFS will help to extend the portfolio of applications for topographic optical imaging in scientific and industrial scenarios involving rapid processes and dynamic samples.

## Methods

### Instrumentation
The measurements in the first prototype were performed using a Nikon 20x/0.45NA microscope objective mounted on a motorised z stage (DOF-5, Dover Motion). The measuring head includes a LED with a central wavelength of 525 nm, a structured illumination channel with a static chrome-on-glass checkerboard pattern of 13 μm pitch that is projected onto the sample, a tube lens with 50 mm focal length and a camera (VCXU-13M, BAUMER) with 1024 × 1280 pixels and 4.8 μm pixel size. Implementation of the illumination sequences driving the LED that generate the encoding patterns, and their synchronisation with the scanning of the z-stage and camera exposure was performed with custom software and an Arduino board (Mega 2560, Arduino).

The measurements of the second prototype were performed with a Nikon Eclipse Ti2 commercial microscope equipped with a Nikon 20x/0.45NA microscope objective. The optical setup included an LED with central wavelength of 525 nm, a structured illumination channel with a static chrome-on-glass Ronchi ruling of 50 μm pitch that is projected onto the sample, a TAG lens (TAG 2.0, TAG Optics Inc.), a pair of relay lenses, a tube lens with 200 mm focal length and a camera (DMK33UX287, the Imaging Source) with 540 × 720 pixels and 6.9 μm pixel size. The TAG lens was driven by an arbitrary waveform generator (SDG6022X, Siglent). Implementation of the LED illumination sequences and their synchronisation with the TAG lens and camera exposure was performed by a field-programable gate array (FPGA) (DE0-Nano, Terasic).

### Samples
Surface roughness measurements were performed on the certified material measure NPL AIRB40, with certified roughness values of Sa = 0.79 μm with expanded uncertainty of 0.03 μm, and Sq = 1.00 μm with expanded uncertainty of 0.02 μm. For calibration and measurement, we employed a series of NPL Areal standard 027 type PGR step height material measure samples[33] with certified values of 1918 nm with expanded uncertainty of 6 nm, and 926 nm with expanded uncertainty of 6 nm; and SiMetricS step height with the certified value of 21.70 μm with expanded uncertainty of 0.04 μm. To perform calibration and measurements of a flat sample we employed an aluminium mirror with λ/10 flatness. Measurements also included regions on a 10 cent euro coin, and a micro hotplate of a gas sensor driven with an electrical signal at 3 Hz and 10 $V_{pp}$. All expanded uncertainties have a coverage factor of 2.

### Reconstruction algorithm
The metric on Eq. (2) is implemented with standard deviations $\sigma_a = 1.5$ and $\sigma_b = 5$ pixels. Each optical system has been previously calibrated

with a flat aluminium mirror to compensate for the field curvature. All topographies were calculated using Python programming language and analysed using MountainsMap (Digital Surf) except for the micro hot-plate video, which was calculated using MATLAB (The MathWorks).

The most time-consuming tasks during topography reconstruction are computing the optically sectioned signal (Eq. (2)) and the arctangent (Eq. (6)). Note that the former is performed independently for each image so it can be parallelised. A straight CPU-based implementation in C + + processing 512 × 512 pixel images using a standard PC required less than 30 ms per reconstruction, suggesting that an optimised implementation including GPU computation would bring the processing time below the acquisition time (15 ms using the second prototype).

## Data availability
The authors declare that the data supporting the findings of this study are available within the paper and its supplementary information files. Source data are provided with this paper.

## Code availability
The procedures for reconstructing ESFS topography maps are provided in the Supplementary Information.

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

## Acknowledgements

The authors would like to acknowledge Prof. Albert Romano for providing the MEMS devices. N.V., G.C. and M.D. would like to thank AGAUR for supporting this project with an Industrial PhD Grant DI-2021. M.D. acknowledges funding from the European Research Council (ERC) under the European Union's Horizon 2020 research and innovation programme (grant agreement No. 101002460). M.D. is a Serra Hunter professor.

## Author contributions

G.C. conceived the research. G.C. and M.D. supervised the research. G.C. and N.V. conceived and designed the analysis. N.V. performed the experiments and analysed the data with support from R.A., G.C. and M.D.. G.C. and M.D. wrote the manuscript with inputs from N.V..

## Competing interests

The authors declare no competing interests.
