## [Peer Review File · Nature Communications]

Fast topographic optical imaging using encoded search focal scanEditorial Note: Parts of this Peer Review File have been redacted as indicated to remove third-party material where no permission to publish could be obtained.

REVIEWER COMMENTS

Reviewer #1 (Remarks to the Author):

This work reported a fast encoded search focal scan technique for topographic optical imaging. Rather than acquiring a series of focal images in sequence which is widely adopted in traditional optical sectioning method, authors proposed to encode Gray code and sinusoidal illumination patterns in different focal depths. In such way, optical sectioning number can be obviously reduced. And authors described the concept and show some results of dynamic measurement using the proposed method. Overall, the idea of this work is ingenious and interesting, but I think the description in current manuscript is a little difficult to understand, which caused some concerns needed to be further clarified. And the specific comments and concerns are as follow:

1. For my understanding, the core idea of this work is to seek a discrete measuring method to represent a continuous signal. And the proposed method reminds me the technique development in fringe projection profilometry (FPP). At first, a series of binary or Gray code structured light patterns are used to modulate and measure surface's height by detecting the deformation of boundary between black and white stripes [1]. So, only the data on several discrete lines can be reconstructed. Then, phase-shifting algorithm plus binary or Gray code patterns [2,3] are proposed and widely used in this field so that the continuous result can be retrieved using much less projected patterns, which is also the key point in this work.
2. Is it required to realize in-plane scanning in each focal plane or just generating full-field structured light illumination? And ronchi pattern is used to generate illumination, and is it fixed? If so, how to switch the encoding pattern from phase shifting sinusoidal patterns to Gray code patterns?
3. I am confused about the projecting sequence in temporal and spatial domains. In the part of implementation, I noticed that the sample moves through the entire measurement range during once exposure of camera, but the illumination/projecting varied in the z-axis

scanning as described in Figs. 1 (b) and 1(g). Therefore, the illumination patterns in different scanning depth are expected to be shown and sequential relationship between illumination and focal sweep should be described more clearly to help audiences better understand this work.

4. A comparative experiment and accuracy evaluation on the standard component using the proposed encoded search focal scan and the traditional focal scan is suggested to be added.

5. Authors mentioned the proposed method allows the real-time measurement of large or rapidly moving samples, so how to eliminate the effect caused by samples' motion during the scanning process (It may cause artifact or multi layers on reconstructed results). In addition, how much time does it cost for once reconstruction, can it be satisfied for real-time measurement?

6. In Supplementary Figure 2, why is it used pulsed illumination with one period of sinusoidal pattern rather than 1/2 period? Now, only the upper part of the sinusoidal pattern is generated after low-pass filtering.

7. More description of Supplementary Figure 8 is required, and the obtained patterns reminds me the two other technique including optical sectioning structured illumination microscopy (SIM) [4,5] and microscopic fringe projection profilometry (FPP) [6]. The former technique can achieve high axial resolution which is equal to that of laser scanning microscopy and the latter does not need axial scanning. Therefore, some discussion of the pros and cons of the proposed method compared with these two techniques are suggested to be added.

Reference

[1] Inokuchi S. Range imaging system for 3-D object recognition[J]. ICPR, 1984, 1984.

[2] Sansoni G, Carocci M, Rodella R. Three-dimensional vision based on a combination of gray-code and phase-shift light projection: analysis and compensation of the systematic errors[J]. Applied optics, 1999, 38(31): 6565-6573.

[3] Wu Z, Guo W, Zhang Q. Two-frequency phase-shifting method vs. Gray-coded-based method in dynamic fringe projection profilometry: A comparative review[J]. Optics and Lasers in Engineering, 2022, 153: 106995.

[4] Neil M A A, Juškaitis R, Wilson T. Method of obtaining optical sectioning by using structured light in a conventional microscope[J]. Optics letters, 1997, 22(24): 1905-1907.

[5] Qian J, Lei M, Dan D, et al. Full-color structured illumination optical sectioning

microscopy[J]. Scientific reports, 2015, 5(1): 14513.

[6] Hu Y, Chen Q, Feng S, et al. Microscopic fringe projection profilometry: A review[J]. Optics and lasers in engineering, 2020, 135: 106192.

Reviewer #2 (Remarks to the Author):

The document discusses a technique for fast topographic optical imaging using an encoded search focal scan (ESFS). The authors claim that this technique allows for significant speed improvements in topographic imaging compared to traditional methods such as confocal microscopy and interferometry. They propose using a binary search approach to determine the axial location of the sample using a set of images captured while the optical focus is scanned through the entire measurement range.

While the concept of ESFS sounds promising, it is important to critically evaluate the claims made in the document. One potential concern is the reliance on a binary search approach. The document mentions that by properly selecting the merged interrogation planes, it should be possible to determine the axial position of the sample using only a logarithmic number of images. However, it is unclear how this selection process is performed and whether it is robust enough to accurately determine the sample's position.

Another point to consider is the trade-off between imaging speed and reconstruction fidelity. The document mentions that the computational approaches used in ESFS, such as compressed sensing and point-spread function engineering, can enable full topographic reconstruction from far fewer images compared to traditional methods. However, it is important to assess the potential loss of reconstruction fidelity that may occur as a result of using these techniques. Additionally, the document mentions that these techniques are computationally expensive and require offline processing, which raises questions about the practicality and real-time applicability of ESFS.

Furthermore, the document discusses the implementation of ESFS in two different prototypes. While the results presented in the document show promising topographic reconstructions of step height samples and flat mirrors, it is important to consider the limitations and potential sources of error in these experiments. For example, the document mentions that the measurements were performed using specific microscope objectives and cameras, and it is unclear how the results would generalize to different optical setups.

Finally, it is not clear to me how this technique relate to standard structured light

profilometry (SLP) techniques? 3D measurements can be done in 3 (or even 1) structured light pattern, so speed can be very high here, what about the resolution comparison? In conclusion, while the concept of fast topographic optical imaging using ESFS is intriguing, there are several aspects that need to be further investigated and validated. The claims made in the document should be critically evaluated, and additional experimental evidence and comparisons with existing techniques would be necessary to fully assess the potential of ESFS in practical applications.D

Reviewer #3 (Remarks to the Author):

This manuscript proposed a novel method called Encoded Search Focal Scan (ESFS) to dramatically reduce the required images in confocal-like systems, which enables real-time topographic reconstruction. The proposed method employs encoded illuminations in focal sweep and decodes the best focal positions with two stages including binary decoding and phase-shifting using the focal sweep images. The proposed method is sound. I believe this work will be impactful the the field. Thus I recommend the acceptance of this paper provided that the authors address minor comments below:

1. The authors claim that the method can achieve topography maps at diffraction-limited resolution, but will the focal sweep involved in the method affect the final resolution? It would be better to show some experimental data to demonstrate the spatial resolution.
2. In the introduction, it could be misleading to claim that the speed will only be limited by the camera frame rate, since the speed of focal sweeping is obviously another restriction of this method. Especially when the lens is mechanically translated such as the experiments conducted in the first prototype system, the focal sweeping speed becomes the main barrier of the speed.
3. Could the authors explain more on the derivation of Equation 5?
4. Please check that all symbols used in the equations have specific explanations to avoid ambiguities. For example, the symbol “i” in Equation 1, symbol “T” in Equation 4.

POINT BY POINT ANSWER TO REVIEWERS' COMMENTS

We would like to thank all the reviewers for their accurate work of revision and for their useful comments and suggestions, which helped us to improve the quality of the manuscript. Please, find below a list of point-by-point answers (in blue) to their comments (in black) along with the changes performed in the revised version of the manuscript (in red).

REVIEWER COMMENTS

Reviewer #1 (Remarks to the Author):

This work reported a fast encoded search focal scan technique for topographic optical imaging. Rather than acquiring a series of focal images in sequence which is widely adopted in traditional optical sectioning method, authors proposed to encode Gray code and sinusoidal illumination patterns in different focal depths. In such way, optical sectioning number can be obviously reduced. And authors described the concept and show some results of dynamic measurement using the proposed method. Overall, the idea of this work is ingenious and interesting, but I think the description in current manuscript is a little difficult to understand, which caused some concerns needed to be further clarified. And the specific comments and concerns are as follow:

1. For my understanding, the core idea of this work is to seek a discrete measuring method to represent a continuous signal. And the proposed method reminds me the technique development in fringe projection profilometry (FPP). At first, a series of binary or Gray code structured light patterns are used to modulate and measure surface's height by detecting the deformation of boundary between black and white stripes [1]. So, only the data on several discrete lines can be reconstructed. Then, phase-shifting algorithm plus binary or Gray code patterns [2,3] are proposed and widely used in this field so that the continuous result can be retrieved using much less projected patterns, which is also the key point in this work.

This is an interesting point. The core idea of the work is decreasing the time required to reconstruct the topography of a sample at microscale resolution by reducing the number of necessary images. This has been a long-term quest in optical metrology. Importantly, the problem of fast topographic optical imaging is not restricted to the microscale, but it also affects sample characterization at low spatial resolutions – tens or hundreds of microns. It is in this framework that the technique fringe projection profilometry (FPP), also known as structured projection profilometry (SPP), was created.

As the reviewer points out, recent developments in FPP include a two-step method based on projecting phase-shifting sinusoidal patterns at the surface of a sample followed by a phase unwrapping approach such as using Gray code patterns. This is indeed similar to the approach proposed here, but with a fundamental difference. In our case, we do not generate lateral encoding patterns, as in FPP, but along the optical propagation axis (z-axis). This makes our technique compatible with any system where a high numerical aperture is used, including confocal microscopes and focus variation approaches. Therefore, our technique exhibits a lateral resolution one or two orders of magnitude higher than FPP.

To clarify this aspect, we have added the following paragraph in the main text (line 311):

“This two-step process resembles that used in fringe projection profilometry (FPP) to increase topographic imaging speed³¹. In this case, different patterns of structured light are sequentially

projected at the sample, from which the sample axial position can be inferred by triangulation – no axial scanning is needed. However, a measurement range exceeding the depth-of-field of the detection optics impacts the axial and, severely, the lateral resolutions due to the defocus blurring. Coupled with the geometry constraints of angled projection and limitations of triangulation, FPP typically features a spatial resolution of hundreds of microns, leaving the technique out of the domain of high-resolution microscopy³². Such resolution can be orders of magnitude worse than what is possible using ESFS, where the use of full NA is enabled, and the measurement range can be arbitrarily extended.”

where new Ref. 31 is “Wu Z et al. *Two-frequency phase-shifting method vs. Gray-coded-based method in dynamic fringe projection profilometry: A comparative review*. Optics and Lasers in Engineering, 2022, 153: 106995”, and new Ref. 32 is “Hu Y, et al. *Microscopic fringe projection profilometry: A review*. Optics and lasers in engineering, 2020, 135: 106192”.

2. Is it required to realize in-plane scanning in each focal plane or just generating full-field structured light illumination? And ronchi pattern is used to generate illumination, and is it fixed? If so, how to switch the encoding pattern from phase shifting sinusoidal patterns to Gray code patterns?

In the presented form of the technique, it is not required to perform in-plane scanning: the mechanism with which the optically-sectioned images are obtained is based on projecting a pattern (structured illumination) and detecting whether the pattern is or is not in the images. This is effectively an “active-illumination assisted Focus Variation” method to perform the optical sectioning in which in-plane scanning is not required. Because of this, the lateral resolution is affected to some extent, compared to other optical-sectioning techniques that use in-plane scanning, such as confocal microscopy. We have now clarified this in the manuscript (at line 124):

“...we detect the presence or absence of the sample at any of the locations masked axially by the binary modulation $M_i(z)$. Note how this calculation of the focus-sensitive signal involves spatial filtering on a single image and, although this affects the lateral resolution to some extent, it does not require in-plane scanning.”

As per the second question, precisely because we chose structured illumination combined with a Focus Variation operator to perform the optical sectioning, the pattern that we project (based on a checkerboard pattern in the first prototype, and on a Ronchi pattern for the second prototype) is fixed and the Focus Variation operator uses information in the neighbourhood of each pixel to determine the optical-sectioning signal. We believe this is now clarified with the addition of the previous text, and we have also explicitly specified that the pattern is fixed (line 202):

“They both feature a pulsed Light Emitting Diode (LED) for the light modulation, and a fixed patterned amplitude mask to generate the structured illumination...”

and (line 372):

“The measuring head includes a LED with a central wavelength of 525 nm, a structured illumination channel with a static chrome-on-glass checkerboard pattern...”

and (line 381):

“The optical setup included an LED with central wavelength of 525 nm, a structured illumination channel with a static chrome-on-glass Ronchi ruling...”

Finally, the fixed pattern used for the structured illumination should not be confused with the encoding patterns, which occur along the optical axis (z-axis). Specifically, we axially translate the microscope objective with respect to the sample (first prototype) or the focal plane (second prototype) while modulating the illumination in time. Thus, the pattern is fixed but the illumination intensity varies with time (either on-off, sinusoidally, or pulsed). We achieve this by implementing a fixed transmissive mask in the illumination path and controlling the illumination intensity by modulating the driving of the LED. We have now emphasised this in the manuscript (line 375):

“...with 1024×1280 pixels and $4.8 \mu\text{m}$ pixel size. **Implementation of the illumination sequences driving the LED that generate the encoding patterns, and their synchronisation with the scanning of the z-stage and camera exposure** was performed with custom software and an Arduino board (Mega 2560, Arduino).”

and (line 385):

“The TAG lens was driven by an arbitrary waveform generator (SDG6022X, Siglent). **Implementation of the LED illumination sequences and their synchronisation with the TAG lens and camera exposure** was performed by a field-programable gate array (FPGA) (DE0-Nano, Terasic).”

3. I am confused about the projecting sequence in temporal and spatial domains. In the part of implementation, I noticed that the sample moves through the entire measurement range during once exposure of camera, but the illumination/projecting varied in the z-axis scanning as described in Figs. 1 (b) and 1(g). Therefore, the illumination patterns in different scanning depth are expected to be shown and sequential relationship between illumination and focal sweep should be described more clearly to help audiences better understand this work.

The reviewer is correct in the understanding that the focal plane is moved through the entire measurement range during one exposure of the camera, and that the illumination/projecting is varied through the z-axis scanning, as indicated in Figure 1(d) and 1(g). However, as noted also in the previous point, the spatial pattern is fixed and what is varied during the z-axis scanning is the illumination intensity. We believe that the additions highlighted in the previous point make the difference between the spatial-domain pattern (fixed) and the temporal-domain pattern (modulated in time) clearer. Also, we have now further clarified the relationship between illumination and focal sweep, in the described example of one illumination sequence we have added (line 129):

“A simple example of the information extracted from a focal-sweep image is the implementation $M_i(z) = 1$ for $z < (z_{\text{max}} - z_{\text{min}})/2$ and $M_i(z) = 0$ otherwise. **That is, the illumination would be on during the first half of the exposure time and off during the second half, making the fixed structured illumination pattern only visible if the sample is in-focus at some point during the first half of the exposure time. Therefore, in this example sequence,** a value of $S_i(x, y)$ above (or below) the set threshold would indicate that the height of the sample at (x, y) is somewhere in the first (or second) half of the measurement range.”

Additionally, since this relationship is further described more in depth in Supplementary Note 1, we have now indicated this (line 136):

“...The combination of a few images with an appropriately designed set of sequences $\{M_i(z)\}$ can be used to perform an axial search of the height at each pixel (see Supplementary Note 1)...”

4. A comparative experiment and accuracy evaluation on the standard component using the proposed encoded search focal scan and the traditional focal scan is suggested to be added.

We have added experiments using standard components to assess the accuracy and to provide a comparison with a traditional focal scan, including an assessment of the spatial resolution. First, we have added measurements on a topographic Siemens star target, which provide an indication of the resolution achieved. We have added the following paragraphs in Supplementary Note 4:

“To assess the lateral resolution of the system, and to compare it with a conventional method for reference, we measured a topographic Siemens star target, as shown in Supplementary Figure 9. The target consists of 18 pairs of circular sectors with a nominal height of $0.186\mu\text{m}$. We performed two different sets of measurements both using a $50\times/0.8\text{NA}$ objective and a total measurement range of $15\mu\text{m}$. The first one with ESFS and the second with a conventional technique for comparison. The selected technique is "focus variation with structured illumination" as it provides a fair comparison with our implementation of ESFS. For each case, the measured height was calculated by analysing the contrast of concentric circular profiles. Each profile (at varying radial distances) samples the modulated height of the target at a different spatial frequency. This enables to compute the Instrument Transfer Function (ITF) of the system, a quantity that relates the ratio between the measured to the nominal height as a function of the spatial frequency. The cut-off frequency of the ITF is therefore an indication of the topographic resolution of the system. As shown in Supplementary Figure 9(b), the calculated ITF in the case of the conventional method reaches higher spatial frequencies than ESFS. This indicates a loss of spatial resolution in the latter, arguably caused by the lower SNR in the focal-sweep images.

Given that the current ESFS implementation is based on focus variation, the reported lateral resolution values depend on the kernel filters applied (σ_b in the outer filter in Equation (2)). Increasing σ_b reduces the lateral resolution, but a minimum value of this parameter is necessary to calculate the optical section image needed for topography reconstruction. Instead, and as stated before, the system noise decreases with σ_b . To experimentally quantify this trade-off for the case of ESFS and also conventional focus variation, we calculated the loss in lateral resolution and the reduction in system noise as a function of σ_b . The resolution was estimated as the inverse of the spatial frequency at which the ITF falls at 10%, and the system noise was computed by subtracting two consecutive measurements divided by $\sqrt{2}$. The results shown in Supplementary Figure 9 are in good agreement with the expected behaviour. Interestingly, the system noise is significantly increased in the case of ESFS compared to traditional focus variation, but the lateral resolution is only slightly sacrificed. Importantly, though, the number of images required to properly reconstruct a topography map is drastically reduced in ESFS, as widely discussed in the main text.”

and we added the new Supplementary Figure 9, with the results.

Additionally, we have added a summary of results of measurements performed on standard targets (AirB40 specimen from NPL and several certified step heights) in the new Supplementary Table 3.

Finally, to emphasise these results we have added the following in the manuscript (in line 234):

“...Further details of the characterisation of the system can be found in Supplementary Information, including a comparative analysis of the topographic spatial resolution and system noise achieved with ESFS and with a conventional technique based on plane-by-plane scanning. Results show that ESFS leads to increased system noise but causes only a slight reduction in the lateral resolution.”

5. Authors mentioned the proposed method allows the real-time measurement of large or rapidly moving samples, so how to eliminate the effect caused by samples' motion during the scanning process (It may cause artifact or multi layers on reconstructed results). In addition, how much time does it cost for once reconstruction, can it be satisfied for real-time measurement?

This is a pertinent point. We have now added the following discussion (line 323):

“As with any measurement technique, motion artifacts would be expected if one attempts to capture dynamic processes that change faster than the sampling rate. However, the high reduction in the number of input images increases the sampling rate, effectively rendering ESFS as a real-time technique. For instance, in our implementation in the second prototype, data acquisition time for one measurement was only 15ms, enabling imaging at 67 topographies per second. Furthermore, reconstruction in ESFS is not a computationally intensive operation, readily enabling online processing at real-time, see Methods.”

As per the second point regarding reconstruction time, we have now added (line 408):

“The reconstruction time is dedicated mainly to the computation of the optically-sectioned signal (Equation 2) and the arctangent (Equation 6). Note that the former is performed independently for each image so it can be parallelised. A straight CPU-based implementation in C++ processing 512×512 pixel images using a standard PC required less than 30ms per reconstruction, suggesting that an optimised implementation including GPU computation would bring the processing time below the acquisition time (15ms using the second prototype).”

6. In Supplementary Figure 2, why is it used pulsed illumination with one period of sinusoidal pattern rather than 1/2 period? Now, only the upper part of the sinusoidal pattern is generated after low-pass filtering.

The period is fixed by design as the step size of the first stage of the technique. In the figure, we compare, for the second stage, sinusoidal illumination vs pulsed illumination, but in both cases the period must match the set period T .

The reviewer is correct in that after low-pass filtering the pulsed illumination resembles a function that appears to generate “only” the upper part of the sinusoidal pattern. Interestingly, though, this function (with period T) can be used for calculating the axial location of the sample. This is precisely what is shown in the figure: even in the case that the generated signal deviates from the sinusoidal function (as is the case, pointed out by the reviewer), the reconstruction results show minimal deviation. This is shown in the bottom-right graph of the figure, where results following from sinusoidal illumination and pulsed illumination are compared.

We believe this is properly described in Supplementary Note 1.1. But we agree that referencing the Note at the main text would help. We have therefore added (in line 217):

“...For the second stage, we implement pulsed illumination with a matching period of 12.5ms, and acquire 4 images with equally-distributed phase shifts (in practice, implementing pulsed illumination with period T provides virtually identical results as implementing sinusoidal modulation, see Supplementary Note 1)...”

7. More description of Supplementary Figure 8 is required, and the obtained patterns reminds me the two other technique including optical sectioning structured illumination microscopy (SIM) [4,5] and microscopic fringe projection profilometry (FPP) [6]. The former technique can achieve high axial resolution which is equal to that of laser scanning microscopy and the latter does not need axial scanning. Therefore, some discussion of the pros and cons of the proposed method compared with these two techniques are suggested to be added.

We agree that the old Supplementary Figure 8 (now Supplementary Figure 4) requires more description. We have expanded its caption to include:

“Processing pipeline of the ESFS method for the reconstruction of the topography of a tilted mirror. Eight input images are shown at the top, four for the first stage (left) and four for the second stage (right). From these images, calculation of the focus-sensitive signal is computed using Equation (2) of the main text, and the resultant images are shown in the second row. For the first stage (left), focus-sensitive images are binarised and a coarse/discretised topography map is calculated. For the second stage (right), the phase-shifted focus-sensitive signals are used to calculate the more precise but wrapped topography of the sample. Finally, the coarse and wrapped components are combined to yield the output topography (bottom-right).”

Regarding the possible resemblance between the patterns displayed in Supplementary Figure 8 (now Supplementary Figure 4) and those used in SIM or FPP, we would like to note that it is just a coincidence. In ESFS, we perform active illumination while axially scanning the focal plane. Such illumination is performed based on a coding sequence, and in the new Supplementary Figure 4, we can observe the frames captured when applying ESFS for the particular case of a titled mirror. Therefore, the stripes observed are due to the axial coding sequence applied. In other words, the stripes represent the experimental result of the code shown in Figure 1d.

As the reviewer points out, both SIM and FPP can be used to extract the topography of a sample. While SIM can provide high axial resolution, it is necessary to collect a z-stack with a minimum of 3 images per plane in which a projected pattern is laterally displaced. Importantly, though, ESFS could be applied with SIM or laser scanning confocal microscopy, given that these techniques already provide a focus-sensitive signal – the only requirement to ensure that the axial coding sequence proposed works. We simply opted to prove its working principle with focus variation because only one image per plane is required without in-plane scanning. We have clarified this point by adding the following description in the results section of the main text (line 108):

“Such focus-sensitive signal can be obtained, for instance, in confocal systems by using a pinhole²¹ or in structured illumination microscopy²², but in-plane scanning or multiple images per plane are needed, respectively. Alternatively, it is possible to achieve a focus-sensitive signal by simply projecting a static illumination pattern on the sample and analysing the high spatial-frequency content from a single captured image – without in-plane scanning.”

Where new Ref 21 is “Pawley, J. Handbook of Biological Confocal Microscopy. (Springer, 2006)”, and new Ref 22 is “Neil, M. A. A., Juškaitis, R. & Wilson, T. Method of obtaining optical sectioning by using structured light in a conventional microscope. Opt Lett 22, 1905 (1997)”

Concerning FPP, we have added a new paragraph in the main text to discuss its differences with ESPS, as detailed in the answer to the first comment.

Reviewer #2 (Remarks to the Author):

The document discusses a technique for fast topographic optical imaging using an encoded search focal scan (ESFS). The authors claim that this technique allows for significant speed improvements in topographic imaging compared to traditional methods such as confocal microscopy and interferometry. They propose using a binary search approach to determine the axial location of the sample using a set of images captured while the optical focus is scanned through the entire measurement range.

1- While the concept of ESFS sounds promising, it is important to critically evaluate the claims made in the document. One potential concern is the reliance on a binary search approach. The document mentions that by properly selecting the merged interrogation planes, it should be possible to determine the axial position of the sample using only a logarithmic number of images. However, it is unclear how this selection process is performed and whether it is robust enough to accurately determine the sample's position.

We thank the reviewer for the positive evaluation of our work. As described below, we have added information in the main text to clarify how the axial illumination encoding, consisting of a Gray code and sinusoidal illumination, is performed. Such encoding determines the interrogation planes. We have clarified that through ESFS a binary search is possible, and it enables unequivocally (i.e robustly) to determine the step number with only $\log_2(N)$ images (in line 84):

“... Indeed, by properly selecting the merged interrogation planes, a binary search is possible, enabling to determine the axial position of the sample using only $\log_2(N)$ images.”

Also, we have clarified how the selection process (of the merged interrogation planes) takes place (in line 92):

“... During the focal sweep, the illumination is turned on and off with a precisely controlled sequence that is different and unique for each image. We denote a particular illumination sequence $M_i(z)$ where z is the continuous axial coordinate, such that $M_i(z) = 0$ implies the illumination is off at plane z and $M_i(z) = 1$ implies the illumination is on. Therefore, setting $M_i(z)$ as an on-off sequence, effectively determines the groups of merged interrogation planes. For a given $M_i(z)$, the corresponding acquired image can be written as,”

Additionally, as a minor modification we have opted to update the following sentence (in line 212), because even if we acquire an additional image (as extensively explained in Supplementary Information, and also mentioned later in the main paper in line 220-221) it is indeed clearer in the local context with the update:

“...Note that the measurement range is approximately 40 times the depth-of-field, so the maximum number of binary-modulated images would be 6 (as is the lowest integer satisfying $2^n > 40$)...”

Finally, we have also performed additional experiments, including the measurements of a Siemens star and different step-heights, to evaluate the robustness of our method. These are included in the new Supplementary Figure 9 and the new Supplementary Table 3.

2- Another point to consider is the trade-off between imaging speed and reconstruction fidelity. The document mentions that the computational approaches used in ESFS, such as compressed sensing and point-spread function engineering, can enable full topographic reconstruction from far fewer images compared to traditional methods. However, it is important to assess the potential loss of reconstruction fidelity that may occur as a result of using these techniques. Additionally, the document mentions that these techniques are computationally expensive and require offline processing, which raises questions about the practicality and real-time applicability of ESFS.

We would first like to clarify that ESFS does not make use of compressed sensing or point-spread engineering. The gist of ESFS is to reduce the number of acquired images by performing a clever interrogation of the sample based on axial scanning and active illumination. The inverse problem of finding the surface topography from the reduced data set collected is well-posed, robust, and not computationally expensive. As such, ESFS cannot be categorized as a computational approach, obviating the problems that these methods suffer, and offering a ready solution to fast topographic imaging at the microscale.

We have added additional information in the introduction (line 62), to elucidate this point:

“As a result, the strong data sparsity inherent in three-dimensional topographic imaging can be exploited, enabling order-of-magnitude improvements in acquisition, **only limited by camera frame rate and axial sweeping time, without added computational complexity or sacrifice in reconstruction fidelity.**”

Regarding fidelity, we performed a more in-depth analysis of the performance of ESFS. To this end, we measured a Siemens star and different step-heights, as described in the answer to comment#5.

3- Furthermore, the document discusses the implementation of ESFS in two different prototypes. While the results presented in the document show promising topographic reconstructions of step height samples and flat mirrors, it is important to consider the limitations and potential sources of error in these experiments. For example, the document mentions that the measurements were performed using specific microscope objectives and cameras, and it is unclear how the results would generalize to different optical setups.

The sources of error in ESFS are mainly two: (a) zero-mean errors due to noise in the acquired images, and (b) 2π errors due to the misclassification of pixels in their step number. These two sources of error are originated in stages two and one of ESFS, respectively. Since these are fundamental errors arising from a finite SNR in the acquired images, it is important to evaluate them quantitatively. This is done in Supplementary Note 3 using simulated data. Additionally, we have now included an experimental assessment of the error (in terms of localisation precision) as a function of the parameter σ_b (blur strength as defined in the outer filter in Equation (2)), which is included in the new Supplementary Figure 9, in graphs (d) and (e). The sources of error also affect the topographic lateral resolution of a system implementing ESFS, and this is also now

assessed by imaging a topographic Siemens star target. Results are shown in the new Supplementary Figure 9 (a-c).

Besides 2π errors, which are virtually suppressed in the reconstruction approach described in Supplementary Note 1, there are no significant sources of error other than detection noise, as in any traditional technique.

As per the second point, ESFS is absolutely generalisable to any optical setup, in which it is possible to include the components required to implement ESFS: a means to modulate the global illumination light, a means to extract a signal that is sensitive to focus (detection of a fixed projected pattern in our case), and a means to perform a focal scan synchronised with the exposure time of the camera. We believe this is well explained in the manuscript. Nonetheless, the new addition with results shown in the new Supplementary Figure 9, and those shown in Supplementary Table 3, are acquired using the first prototype with two different microscope objectives, and also using the second prototype. We believe this provides sufficient confidence that the results are generalisable to any optical setup.

4- Finally, it is not clear to me how this technique relate to standard structured light profilometry (SLP) techniques? 3D measurements can be done in 3 (or even 1) structured light pattern, so speed can be very high here, what about the resolution comparison?

The technique presented here is suitable for confocal-like and focus-variation systems characterized by relatively high numerical aperture optics, which allows for topographic optical imaging at the micro-scale - spatial resolution of some microns or below. These systems obtain a topography map by acquiring a sequence of images at different focal planes. Therefore, the focus needs to be axially translated across the sample.

Instead, structured light profilometry (SLP), also known as fringe projection profilometry (FPP), is based on projecting different patterns of structured light on the sample, from which the sample axial position can be inferred by triangulation – no axial scanning is needed. Because fast projection of light patterns can be achieved with state-of-the-art digital micromirror devices or LED projectors, FPP can be indeed very fast – several tens of topography maps per second. Also, a binary search scan has also been implemented in FPP for further boosting imaging speed. However, FPP requires the sample to lie within the depth-of-field of the detection system. Coupled with the geometry constrains of angled projection and limitations of triangulation, FPP typically features a spatial resolution of hundreds of microns, orders of magnitude worse than ESFS.

To clarify this point, we have added the following paragraph in the discussion section (line 311):

“This two-step process resembles that used in fringe projection profilometry (FPP) to increase topographic imaging speed³¹. In this case, different patterns of structured light are sequentially projected at the sample, from which the sample axial position can be inferred by triangulation – no axial scanning is needed. However, a measurement range exceeding the depth-of-field of the detection optics impacts the axial and, severely, the lateral resolutions due to the defocus blurring. Coupled with the geometry constrains of angled projection and limitations of triangulation, FPP typically features a spatial resolution of hundreds of microns, leaving the technique out of the domain of high-resolution microscopy³². Such resolution can be orders of

magnitude worse than what is possible using ESFS, where the use of full NA is enabled, and the measurement range can be arbitrarily extended.”

where new Ref. 31 is “Wu Z et al. *Two-frequency phase-shifting method vs. Gray-coded-based method in dynamic fringe projection profilometry: A comparative review*. Optics and Lasers in Engineering, 2022, 153: 106995”, and new Ref. 32 is “Hu Y, et al. *Microscopic fringe projection profilometry: A review*. Optics and lasers in engineering, 2020, 135: 106192”.

5- In conclusion, while the concept of fast topographic optical imaging using ESFS is intriguing, there are several aspects that need to be further investigated and validated. The claims made in the document should be critically evaluated, and additional experimental evidence and comparisons with existing techniques would be necessary to fully assess the potential of ESFS in practical applications.D

Following the reviewer’s suggestions, we performed additional experiments to illustrate the advantages and potential limitations of ESFS. In particular, we imaged a Siemens star with a nominal height of 0.186 μm using prototype 1, a 50x/0.8NA objective, and an axial scanning distance of 15 μm . This experiment allowed us to characterize the instrument transfer function (ITF) of our system, a quantity that relates the ratio between the measured to the nominal height as a function of the spatial frequency. ITF is normally used as an indicator of the spatial resolution of a topographic imaging system. We compared the results obtained with ESFS with those of a conventional focus variation system. As shown in the new supplementary Figure 9, the gain in topographic imaging speed of ESFS results in a loss of spatial resolution and increased system noise. This is expected, given the loss in SNR when integrating multiple illumination planes, but could be significantly improved by using a different focus-sensitive system, such as a confocal with a pinhole.

To clarify this point, we have added the following paragraph in Supplementary Note 4:

“To assess the lateral resolution of the system, and to compare it with a conventional method for reference, we measured a topographic Siemens star target, as shown in **Supplementary Figure 9**. The target consists of 18 pairs of circular sectors with a nominal height of 0.186 μm . We performed two different sets of measurements both using a 50x/0.8NA objective and a total measurement range of 15 μm . The first one with ESFS and the second with a conventional technique for comparison. The selected technique is "focus variation with structured illumination" as it provides a fair comparison with our implementation of ESFS. For each case, the measured height was calculated by analysing the contrast of concentric circular profiles. Each profile (at varying radial distances) samples the modulated height of the target at a different spatial frequency. This enables to compute the Instrument Transfer Function (ITF) of the system, a quantity that relates the ratio between the measured to the nominal height as a function of the spatial frequency. The cut-off frequency of the ITF is therefore an indication of the topographic resolution of the system. As shown in **Supplementary Figure 9(b)**, the calculated ITF in the case of the conventional method reaches higher spatial frequencies than ESFS. This indicates a loss of spatial resolution in the latter, arguably caused by the lower SNR in the focal-sweep images.

Given that the current ESFS implementation is based on focus variation, the reported lateral resolution values depend on the kernel filters applied (σ_b in the outer filter in Equation (2)). Increasing σ_b reduces the lateral resolution, but a minimum value of this parameter is necessary

to calculate the optical section image needed for topography reconstruction. Instead, and as stated before, the system noise decreases with σ_b . To experimentally quantify this trade-off for the case of ESFS and also conventional focus variation, we calculated the loss in lateral resolution and the reduction in system noise as a function of σ_b . The resolution was estimated as the inverse of the spatial frequency at which the ITF falls at 10%, and the system noise was computed by subtracting two consecutive measurements divided by $\sqrt{2}$. The results shown in **Supplementary Figure 9** are in good agreement with the expected behaviour. Interestingly, the system noise is significantly increased in the case of ESFS compared to traditional focus variation, but the lateral resolution is only slightly sacrificed. Importantly, though, the number of images required to properly reconstruct a topography map is drastically reduced in ESFS, as widely discussed in the main text.”

In addition, we have added a summary of the results of measurements performed on standard targets (AirB40 specimen from NPL and several certified step heights) in the new Supplementary Table 3. This helps illustrate the performance of our two prototypes.

With these new results, we have demonstrated the feasibility of ESFS for topographic reconstruction in different experimental sets, including the characterization of different step heights, a Siemens star, a tilted mirror, a coin, a roughness standard, and a vibrating membrane.

Reviewer #3 (Remarks to the Author):

This manuscript proposed a novel method called Encoded Search Focal Scan (ESFS) to dramatically reduce the required images in confocal-like systems, which enables real-time topographic reconstruction. The proposed method employs encoded illuminations in focal sweep and decodes the best focal positions with two stages including binary decoding and phase-shifting using the focal sweep images. The proposed method is sound. I believe this work will be impactful the the field. Thus I recommend the acceptance of this paper provided that the authors address minor comments below:

1. The authors claim that the method can achieve topography maps at diffraction-limited resolution, but will the focal sweep involved in the method affect the final resolution? It would be better to show some experimental data to demonstrate the spatial resolution.

This is a pertinent point. The claim that the topography maps achieve diffraction-limited is an over simplification and needs clarification. Imaging (that is, the acquisition of the input images) is performed at diffraction-limited resolution, besides a reduction in contrast and signal-to-noise ratio due to the focal sweep involved, as the reviewer correctly asserts. Reducing the signal-to-noise ratio effectively reduces the imaging resolution due to noise in the field of imaging. For topographic 3D imaging, assessment of spatial resolution is more involved, because the reconstruction procedure must be included (both in ESFS and in conventional approaches). To assess this experimentally, we have added experiments to directly evaluate the spatial resolution of our system. These are now included in the new Supplementary Note 4 and the new Supplementary Figure 9. In particular, we have added the following paragraph in Supplementary Note 4:

“To assess the lateral resolution of the system, and to compare it with a conventional method for reference, we measured a topographic Siemens star target, as shown in **Supplementary Figure 9**. The target consists of 18 pairs of circular sectors with a nominal height of 0.186 μ m. We

performed two different sets of measurements both using a 50x/0.8NA objective and a total measurement range of 15 μ m. The first one with ESFS and the second with a conventional technique for comparison. The selected technique is "focus variation with structured illumination" as it provides a fair comparison with our implementation of ESFS. For each case, the measured height was calculated by analysing the contrast of concentric circular profiles. Each profile (at varying radial distances) samples the modulated height of the target at a different spatial frequency. This enables to compute the Instrument Transfer Function (ITF) of the system, a quantity that relates the ratio between the measured to the nominal height as a function of the spatial frequency. The cut-off frequency of the ITF is therefore an indication of the topographic resolution of the system. As shown in **Supplementary Figure 9(b)**, the calculated ITF in the case of the conventional method reaches higher spatial frequencies than ESFS. This indicates a loss of spatial resolution in the latter, arguably caused by the lower SNR in the focal-sweep images.

Given that the current ESFS implementation is based on focus variation, the reported lateral resolution values depend on the kernel filters applied (σ_b in the outer filter in Equation (2)). Increasing σ_b reduces the lateral resolution, but a minimum value of this parameter is necessary to calculate the optical section image needed for topography reconstruction. Instead, and as stated before, the system noise decreases with σ_b . To experimentally quantify this trade-off for the case of ESFS and also conventional focus variation, we calculated the loss in lateral resolution and the reduction in system noise as a function of σ_b . The resolution was estimated as the inverse of the spatial frequency at which the ITF falls at 10%, and the system noise was computed by subtracting two consecutive measurements divided by $\sqrt{2}$. The results shown in **Supplementary Figure 9** are in good agreement with the expected behaviour. Interestingly, the system noise is significantly increased in the case of ESFS compared to traditional focus variation, but the lateral resolution is only slightly sacrificed. Importantly, though, the number of images required to properly reconstruct a topography map is drastically reduced in ESFS, as widely discussed in the main text."

Also, to avoid confusion, we have updated the following sentence to simply convey that ESFS is a microscopic technique (in line 56):

"Here, we introduce Encoded Search Focal Scan (ESFS), a new technique that fills this void and enables real-time reconstruction of topography maps at **micro and nano scale resolution resolution...**"

2. In the introduction, it could be misleading to claim that the speed will only be limited by the camera frame rate, since the speed of focal sweeping is obviously another restriction of this method. Especially when the lens is mechanically translated such as the experiments conducted in the first prototype system, the focal sweeping speed becomes the main barrier of the speed.

We agree with the reviewer that the speed limit of ESFS is generally limited by the camera frame rate and focal sweeping time. However, when using fast axial sweeping, as demonstrated in the second prototype, camera frame rate is the only speed limiting factor, as claimed in the abstract.

To clarify this point, we have added the following sentence in the introduction (line 62):

"As a result, the strong data sparsity inherent in three-dimensional topographic imaging can be exploited, enabling order-of-magnitude improvements in acquisition time, **only limited by**

camera frame rate and axial sweeping time, without added computational complexity or sacrifice in reconstruction fidelity.”

3. Could the authors explain more on the derivation of Equation 5?

Inserting Equation (1) into (2) of the main text, we have:

$$S_j(x, y) = \mathcal{G}_{\sigma_b} \{ |\nabla^2 \mathcal{G}_{\sigma_a} \{ I_j(x, y) \} | \} \quad (2)$$

$$= \mathcal{G}_{\sigma_b} \left\{ \left| \int_{z_{\min}}^{z_{\max}} K(x, y, z_s - z) M_j(z) dz \right| \right\} \quad (R1)$$

where we have defined:

$$K(x, y, z) = \mathcal{L}(x, y) * [r(x, y) \cdot L(x, y)] * \text{PSF}(x, y, z) \quad (R2)$$

where r and L are the reflectivity and projected texture at the sample, and $\mathcal{L}(x, y)$ is a Laplacian-of-Gaussian kernel that implements the inner Gaussian filter and Laplacian operators included in Equation (2). Recall that $*$ denotes convolution in the (x, y) plane.

Due to the Laplacian operator, the magnitude of K decays rapidly with z . This is because as defocus increases, the point-spread function becomes bigger and smoother, reducing the output of the Laplacian operator. It can be assumed that $|K|$ vanishes for $z < z_{\min}$ and for $z > z_{\max}$, and therefore the integration limits of Equation (R1) can be safely extended. This enables to consider the integral of Equation (R1) as a convolution (in the axial direction), and we may invoke the convolution theorem to yield:

$$\int_{-\infty}^{\infty} K(x, y, z_s - z) M_j(z) dz = (M_j * K)(x, y, z_s) \quad (R3)$$

$$= \mathcal{F}^{-1} \{ \tilde{K}(x, y, \xi) \cdot \tilde{M}_j(\xi) \} \quad (R4)$$

where $*$ is 1D-convolution and:

$$\tilde{K}(x, y, \xi) = \mathcal{F}\{K(x, y, z)\} \quad (R5)$$

$$\tilde{M}_j(\xi) = \mathcal{F}\{M_j(z)\} \quad (R6)$$

with \mathcal{F} denoting Fourier transform through (only) variable z , and \mathcal{F}^{-1} its inverse.

On one hand, if $M_j(z) = 1$ then,

$$S_j(x, y) = \mathcal{G}_{\sigma_b} \left\{ \left| \int_{-\infty}^{\infty} K(x, y, z_s - z) dz \right| \right\} \quad (R7)$$

$$= \mathcal{G}_{\sigma_b} \{ \mathcal{F}^{-1} [\tilde{K}(x, y, \xi) \delta(\xi)] \} \quad (R8)$$

$$= \mathcal{G}_{\sigma_b} \{ [\tilde{K}(x, y, 0)] \} \quad (R9)$$

$$= S_{\max}(x, y) \quad (R10)$$

where δ is the Dirac delta function and $S_{\max}(x, y)$ is defined as such.

On the other hand, if $M_j(z) = \cos(\omega z)$, we can write,

$$S_j(x, y) = \mathcal{G}_{\sigma_b} \left\{ \left| \mathcal{F}^{-1} \left\{ \tilde{K}(x, y, \xi) \cdot \left(\frac{1}{2} \delta(\xi - \omega) + \frac{1}{2} \delta(\xi + \omega) \right) \right\} \right| \right\} \quad (\text{R11})$$

$$= \mathcal{G}_{\sigma_b} \{ |\tilde{K}(x, y, \omega) \cos(\omega z_s)| \} \quad (\text{R12})$$

where we have exploited that $\tilde{K}(x, y, -\omega) = \tilde{K}(x, y, \omega)$, which holds since $K(x, y, z)$ is real-valued. This result indicates that the output is an attenuated cosine function, which is to be expected for a linear system with a pure frequency as input. In other words, the convolution of a cosine function is also a cosine function albeit with a reduction in contrast. Such contrast reduction is,

$$m = \frac{|\tilde{K}(x, y, \omega)|}{\tilde{K}(x, y, 0)} \quad (\text{R13})$$

Finally, joining both results, for $M_j(z) = \frac{1}{2} \left(1 + \cos \left(\frac{2\pi}{T} z + \delta_j \right) \right)$, we have:

$$S_j(x, y) = \mathcal{G}_{\sigma_b} \left\{ \left| \frac{1}{2} \tilde{K}(x, y, 0) + \frac{1}{2} \tilde{K} \left(x, y, \frac{2\pi}{T} z_s + \delta_j \right) \cos \left(\frac{2\pi}{T} z_s + \delta_j \right) \right| \right\} \quad (\text{R14})$$

$$= \mathcal{G}_{\sigma_b} \left\{ \left| \frac{1}{2} \tilde{K}(x, y, 0) \left(1 + m \cos \left(\frac{2\pi}{T} z_s + \delta_j \right) \right) \right| \right\} \quad (\text{R15})$$

$$= \mathcal{G}_{\sigma_b} \{ |\tilde{K}(x, y, 0)| \} \frac{1}{2} \left(1 + m \cos \left(\frac{2\pi}{T} z_s + \delta_j \right) \right) \quad (\text{R16})$$

$$= S_{\max}(x, y) \frac{1 + m \cos \left(\frac{2\pi}{T} z_s + \delta_j \right)}{2} \quad (5)$$

As a side note, if $K(x, y, z \neq z_s) = 0$, approximating a Dirac delta function, it would approach the limit where there is no contrast reduction, with $m = 1 \forall \xi$, which would correspond to an imaging system with an infinitely small depth-of-field. For a realistic finite depth-of-field imaging system, the transfer function at $\xi = \omega > 0$ would necessarily be $m < 1$.

Finally, we note that if $\text{PSF}(x, y, z)$ is axially symmetric, and therefore so is $K(x, y, z)$, then $\arg\{\tilde{K}(x, y, \omega)\} = 0$, which we have assumed above. If it was not, this would simply add a phase offset to Equation (5).

4. Please check that all symbols used in the equations have specific explanations to avoid ambiguities. For example, the symbol “i” in Equation 1, symbol “T” in Equation 4.

We have added explanation for symbol “i” in Equation (1) (line 100):

“...the operator * denotes two-dimensional convolution in the tangential plane (x, y) , and index i denotes the image/sequence pair within the set of images/sequences”

Also, we have added a reminder of the definition of the symbol “T” closer to Equation (4) for clarity (line 177):

“where δ_j is a phase offset, and T is the step size defined in the previous stage satisfying $TN = Z_{\max} - Z_{\min}$.”

REVIEWERS' COMMENTS

Reviewer #1 (Remarks to the Author):

The authors have addressed the reviewer's concerns, and this manuscript is recommended for publication in Nature Communications.

By the way, with respect to the sixth question in my last comment, the comparative simulation using the traditional sinusoidal signal, the squared signal which is widely used in FPP for binary defocusing projection and the pulsed signal used in this work was performed. The used pulsed signal indeed has higher measurement error. And it is a wise choice.

Figure redacted

Fig. 1. Simulation about discrete binary signal design.

Reviewer #2 (Remarks to the Author):

Thank you for conducting a thorough review of the manuscript. Based on your feedback, I am pleased to endorse it for publication.

Reviewer #3 (Remarks to the Author):

Thank you for addressing my concerns.